# Perinatal Anxiety Symptoms: Rates and Risk Factors in Mexican Women

**DOI:** 10.3390/ijerph18010082

**Published:** 2020-12-24

**Authors:** Janeth Juarez Padilla, Sandraluz Lara-Cinisomo, Laura Navarrete, Ma. Asunción Lara

**Affiliations:** 1Department of Kinesiology and Community Health, College of Applied Health Sciences, University of Illinois at Urbana-Champaign, Champaign, IL 61820, USA; janethj2@illinois.edu (J.J.P.); laracini@illinois.edu (S.L.-C.); 2Innovation and Global Health Department, Ramon de la Fuente Muñiz National Institute of Psychiatry, Tlalpan, Mexico City 14370, Mexico; laurae@imp.edu.mx

**Keywords:** perinatal anxiety symptoms, traditional female role, risk factors

## Abstract

Anxiety during pregnancy and after childbirth can have negative consequences for a woman and her baby. Despite growing interest in the perinatal mental health of Mexican women living in the U.S., perinatal anxiety symptom (PAS) rates and risk factors have yet to be established for women in Mexico. We sought to determine PAS rates and identify risk factors, including the traditional female role (TFR) in a sample of Mexican women. This secondary data analysis is based on 234 Mexican women who participated in a longitudinal study on perinatal depression in Mexico. Anxiety symptoms were assessed in pregnancy and at six weeks postpartum. Rates were determined through frequencies, and multiple logistics regressions were conducted to identify risk factors in the sample. The PAS rate was 21% in pregnancy and 18% postpartum. Stressful life events and depressive symptoms were associated with a higher probability of PAS. Adherence to TFR increased the probability of prenatal anxiety; lower educational attainment and low social support during pregnancy increased the probability of postpartum anxiety. The PAS rates were within the range reported in the literature. The TFR was only associated with anxiety in gestation, highlighting the role of this culturally relevant risk factor. Culturally responsive early interventions are therefore required.

## 1. Introduction

Anxiety during the perinatal period (pregnancy and first postpartum year) is a global health issue because of its negative consequences for the mother and her baby [1,2,3,4]. However, perinatal anxiety symptoms (PAS) are not always diagnosed or treated, increasing the risks for both. Previous data show that women in low- and middle-income countries have a high prevalence of anxiety compared to those in more affluent countries [5]. Although anxiety has been assessed in individuals of Mexican descent living in the U.S. and Mexican women [6,7,8], the prevalence of PAS in Mexico has not been fully studied. To our knowledge, only one study has explored anxiety symptoms in Mexican women. Navarrete et al. (2012) found a 14.8% prevalence of anxiety in pregnancy and 10.6% at six weeks postpartum in Mexican women with depressive symptoms [9]. However, the study did not explore risk factors to explain observed rates because its focus was depressive symptoms.

While risk factors, such as not having a partner [10,11], lower social support [12,13,14], lower partner satisfaction [15,16,17], and multiparity [18] for perinatal depression have been identified, their associations with PAS have yet to be determined among perinatal women living in Mexico. Cesarean delivery and stressful life events [18,19,20,21] are risk factors for PAS in Colombia, Australia, and Rwanda, while resilience has been identified as a protective factor for PAS [22]. Nevertheless, these factors have not been tested in women living in Mexico.

During the perinatal period, Mexican women may experience culturally relevant stressors that may increase their risk of PAS such as traditional female role (TFR) expectations. Adherence to the TFR, defined as being submissive, conforming, indecisive, and passive, attributed to socialization within a structure that undervalues women, is particularly relevant here [23,24]. Although the TFR has been associated with depression in Mexican women at six weeks postpartum [25], its links with anxiety have not been tested. It is important to understand the role of the TFR as a risk factor for PAS to develop interventions and improve women’s mental health during pregnancy and the postpartum period.

The purpose of this study was to determine PAS rates in a sample of women receiving prenatal care who were followed postpartum to identify demographic, psychosocial, and culturally relevant risk factors to inform interventions and treatment options. We hypothesized that (a) women would have higher rates of anxiety during pregnancy than in the postpartum period; (b) cesarean delivery, psychosocial stressors (i.e., high levels of stressful life events, low social support, low educational attainment, not having a partner, and low partner satisfaction) would increase the likelihood of PAS; and (c) high depressive symptoms, low resilience levels, and high adherence to the TFR would increase the likelihood of anxiety during the perinatal period.

## 2. Materials and Methods

### 2.1. Participants

A non-probabilistic sample of 280 pregnant women receiving prenatal care agreed to participate in the parent study. Of these women, 234 (83.6%) were assessed during pregnancy and followed up at six weeks postpartum and were included in this study (Figure 1). Women were approached in the waiting rooms of a hospital that provides comprehensive medical care to state employees and a primary health center that provides prenatal and other medical care to the local community. To be eligible, women had to be at least 20 years of age, ≥26 weeks pregnant, and living in the Mexico City metropolitan area; women with bipolar disorder were excluded. Eligible participants who agreed to be interviewed provided written consent.

### 2.2. Procedures

Data were collected by trained final year psychology students and graduate psychology research assistants affiliated to the lead research institution. Interviews were conducted at ≥26 weeks of pregnancy and six weeks postpartum. The study was approved by the Institutional Review Board (IRB) of Ramon de la Fuente Muñiz National Institute of Psychiatry.

### 2.3. Measures

Sociodemographic information, including age, monthly family income, marital status, and educational attainment, were collected during the enrollment interview (≥26 weeks pregnant). Obstetric data included the number of children (collected at ≥26 weeks pregnant) and type of delivery in this pregnancy (collected at six weeks postpartum). Anxiety symptoms were assessed using the 10-item anxiety subscale from the Symptoms Checklist-90 or SCL-90 [26]. Participants reported the degree of distress they had experienced during the previous fortnight using a five-point Likert scale (none = 0; extreme = 4). The scale was validated in Mexico [27,28]. A score of ≥18 was considered an indicator of high anxiety symptomatology [26]. This instrument showed an internal consistency of α = 0.80 in Mexican women and has been used with perinatal women [9,29].

Because depression is treated as a risk factor here, the Patient Health Questionnaire (PHQ-9) was used to assess depressive symptoms [30]. This is a nine-item depression module from the full PHQ, specifically developed for use in primary care. The PHQ-9 has proven its usefulness as an assessment tool for the diagnosis of depression with acceptable reliability, validity, sensitivity, and specificity; a score of ≥10 indicates a risk of depression [31]. The PHQ-9 was validated in Mexican women, showing an internal consistency of 0.89 [32] and estimated for this sample showing adequate validity: pregnancy: α = 0.78 and six weeks postpartum: α = 0.80.

Resilience was measured with the Resilience Inventory (RESI; [33]), designed to measure resilience in Mexican mothers. The RESI comprises seven dimensions: positive attitude, sense of humor, perseverance, religiosity, self-efficacy, optimism and goal orientation. Response options range from 1 (not at all) to 5 (completely), while the total score indicates the degree of resilience. The RESI was validated in this sample demonstrating good internal consistency (α = 0.88) in pregnant women and adequate concurrent validity for prenatal depressive symptoms (*r* = 0.27; *p* = 0.00) and stressful life events (*r* = −0.19; *p* = 0.00) [34]. In the absence of a pre-established cut-off point, the median minus one standard deviation [34] was used to determine high vs. low resilience.

Stressful life events were measured using the Life Events Scale, a brief, 12 item-form including potential stressors (such as illness, financial problems, accident, job loss or intimate partner violence) selected from the 23 original items [35], developed from Holmes and Rahe’s instrument [36], among other scales. These twelve items were considered more pertinent in the perinatal period [37]. The scale evaluates the occurrence of each event over the past six months as well as the perceived degree of stress it produced (0 = event was not present; 1 = present but produced no stress, 2 = produced little stress, 3 = produced moderate stress, 4 = produced great stress). A cut-off point of 13 was calculated as the 75th percentile to determine high vs. low-stress life events.

The Spanish version [38] of the English Postpartum Depression Predictors Inventory-Revised (PDPI-R) social support subscale [39] was used during pregnancy and postpartum. The 12-item scale quantifies the presence of instrumental and emotional support from the woman’s partner, mother, and other family members and friends. Each item is scored using dichotomous responses with a 0 (absent) or 1 (present), creating a possible summed range of scores from 0 to 12, where higher scores indicate greater support. This instrument has been validated in a sample of pregnant women and has shown adequate concurrent validity in postpartum samples [40]. The scale showed adequate reliability (α = 0.84) [40]. In the absence of a pre-established cut-off point, the median minus one standard deviation [40] was used to determine high vs. low social support.

The marital satisfaction subscale was also taken from the Spanish version [38] of the PDPI-R [39]. The three-item scale measures how satisfied women feel about the relationship with their partner or the father of their child. Each item is scored using dichotomous responses with a 0 (absent) or 1 (present), creating a possible summed range of scores from 0 to 3. This scale showed adequate concurrent validity compared with the Edinburgh Postnatal Depression Scale (*r* = 0.39; *p* = 0.001) in a sample of pregnant Mexican women [40]. A cut-off point was determined using the median minus one standard deviation (1) for high vs. low relationship satisfaction.

The TFR was measured in pregnancy through eight questions from the submissive subscale of the Inventario de Masculinidad y Femineidad (IMAFE; [24]). The scale measures the personality traits expected in many cultures, particularly those of Mexican women [24]. Using a 0 (“I’m never or almost never like this”) to 7 (“I’m always or almost always like this”) scale, participants indicated the extent to which each trait describes them, such that higher scores represented higher levels of TFR. The items included were being conformist, submissive, indecisive, having a weak personality, being resigned, cowardly, withdrawn, and shy. This short version demonstrated adequate reliability in this sample (α = 0.83) [25]. The submissive subscale has also shown adequate reliability in other Latin American countries, such as Colombia [41,42]. A cut-off point was determined using the median plus one standard deviation [25], indicating high vs. low adherence to a TFR.

### 2.4. Data Analyses

Frequencies/percentages were calculated for categorical data. The point prevalence of anxiety symptoms was calculated as the percentage of women with anxiety symptoms at each point of assessment (≥26 weeks pregnant; sixth week postpartum). The McNemar test was performed to determine whether there is a difference in the prevalence of anxiety between the two times. To identify risk factors for anxiety symptoms, a bivariate logistic regression analysis was conducted to estimate odds ratios (ORs) with 95% confidence intervals. To identify the PAS risk factors, multivariate logistic regressions were performed. All analyses were performed using the SPSS version 21 (IBM Corp, Armonk, NY, USA).

## 3. Results

### 3.1. Participants

Participants had an average age of 29 (SD = 6.2), with an average monthly income of $498.70 U.S. dollars (SD = 502.3), an average of 12 years of schooling (equivalent to middle school), and had at least one child. Most had a partner and had given birth by cesarean section. Table 1 shows the distributions of perceived social support, experienced stressful life events, and adherence to the TRF. During pregnancy, 20.4% met the criteria for depressive symptoms, and 17.1% met the cutoff at six weeks postpartum.

### 3.2. PAS Prevalence Rates and Risk Factors

PAS rates (SCL-90 ≥ 18) were 22.1% and 17.5% at six weeks postpartum (shown as Table 2). Results from the McNemar analysis showed that there was no significant difference between perinatal periods (McNemar χ^2^ = 1.30; *p* = 0.25).

### 3.3. Regressions to Determine the Effects of Risk and Protective Factors on PAS

The bivariate logistic regressions to determine associations between psychosocial and culturally relevant factors and anxiety symptoms during pregnancy indicated that prenatal women who were single were at more than twice the risk of reporting prenatal anxiety symptoms than women who were in a relationship (OR = 2.59); prenatal women with low resilience were nearly three times more likely to report prenatal anxiety symptoms than women with high resilience (OR = 2.87); prenatal women with high adherence to the TFR were four times as likely to report prenatal anxiety symptoms than women with low adherence to the TFR (OR = 4.07); prenatal women with low relationship satisfaction had a four times higher risk of reporting prenatal anxiety symptoms than women with high marital satisfaction (OR = 4.03); women with low social support had a greater likelihood of prenatal anxiety symptoms than those with high social support (OR = 6.17); women who experienced stressful events were eight times as likely to report prenatal anxiety symptoms than women who did not (OR = 8.40); having depressive symptoms increased the risk nearly sevenfold (OR = 6.63).

The bivariate logistic regressions (Table 3) to determine the association between psychosocial and culturally relevant factors and anxiety symptoms during the sixth week postpartum showed that prenatal women who were single were over twice as likely to report postpartum anxiety symptoms than married women (OR = 2.25); higher educational attainment is associated with a reduction in the probability of reporting postpartum anxiety (OR = 0.89); prenatal women with low resilience had twice the odds of reporting postpartum anxiety symptoms than women with high resilience (OR = 2.24); prenatal women with low marital satisfaction had more than twice the odds of reporting postpartum anxiety symptoms than women with high marital satisfaction (OR = 2.43); prenatal women with low social support had six times the odds of reporting postpartum anxiety symptoms than women with high social support (OR = 6.63); postpartum women with low social support had twice the odds of reporting postpartum anxiety symptoms than women with high social support (OR = 2.85); prenatal women who experienced stressful events had three times the odds of reporting postpartum anxiety symptoms than women who did not (OR = 3.47); postpartum women who experienced stressful events had seven times the odds of reporting postpartum anxiety symptoms than women who did not (OR = 7.39); prenatal women with depressive symptoms had four times the odds of reporting postpartum anxiety symptoms than women who did not (OR = 4.55); and postpartum women with depressive symptoms had eleven times the odds of reporting postpartum anxiety symptoms than women who did not (OR = 11.25).

In the multivariate model of anxiety in pregnancy, the following variables were included as risk factors: age, monthly family income, educational attainment, number of children, marital status, resilience, TFR, marital satisfaction, social support, stressful life events, and depressive symptoms during pregnancy. The model showed that prenatal women with high adherence to the TFR were at twice the risk of experiencing prenatal anxiety symptoms as women with low adherence to the TFR (OR = 2.28). Stressful events increased the risk of PAS fourfold (OR = 4.35), while prenatal women with depressive symptoms were at four times the risk of reporting prenatal anxiety symptoms as women without them (OR = 3.82) (Table 4).

In the multivariate model of anxiety at six weeks postpartum, the following variables were included as risk factors: age, monthly family income, educational attainment, number of children, marital status, cesarean-delivery, resilience, TFR, marital satisfaction, social support, stressful life events, and depressive symptoms during pregnancy, and social support, stressful life events, and depressive symptoms during the postpartum period. The analysis showed that higher educational attainment was associated with a reduction in the probability of reporting postpartum anxiety symptoms (OR = 0.84), while women with low social support during pregnancy had almost five times the risk of reporting postpartum anxiety symptoms as women with high social support (OR = 5.60). Additionally, the risk of reporting postpartum anxiety symptoms was more than five times higher in those who experienced stressful events post-childbirth than in those who did not (OR = 5.42), while the risk of reporting postpartum anxiety symptoms was nine times higher in those with postpartum depressive symptoms than in those without them (OR = 9.03) (Table 5).

## 4. Discussion

As hypothesized, low levels of social support, stressful life events, and depressive symptoms significantly increased the odds of PAS. The TFR increased the odds of anxiety symptoms during pregnancy. We found that women who reported low levels of social support during pregnancy had five times the odds of PAS. These findings support the growing body of research showing that low social support is associated with increased anxiety during the perinatal period [43,44,45]. However, previous studies did not include women living in Mexico, where there are 34.9 million women of reproductive age [46], with much of the research focusing on English-speaking countries (such as the United States, Europe, and Australia). To our knowledge, this is the first study to assess the role of social support in PAS in Mexico. Given that anxiety might be more culturally acceptable than depression [47], these findings can help pave the way to increasing discussions about perinatal mental health in Mexico, which continues to be severely understudied and inadequately addressed. These findings can also help inform intervention in Mexico and other Latin American countries, where the role of social support during the perinatal period may change as a result of family migration and women’s increased workforce participation [48].

The results also showed that women who experienced stressful life events had nearly four times the odds of experiencing anxiety in pregnancy and nearly five times the odds of experiencing it postpartum. Although the assessment of stressful life events is based on retrospective data, the degree to which they caused stress is informative because stressful life events can have lasting, potentially detrimental effects particularly during a vulnerable period such as pregnancy and first-time motherhood [44,49,50]. Practitioners should therefore assess women living in Mexico for stressful life events to identify women at risk of developing PAS. Researchers are encouraged to develop early interventions to reduce the effects of stressful life events on the development of anxiety in these women.

As predicted, experiencing depressive symptoms at the time of assessment significantly increased the odds of PAS by a factor of 3.82 in pregnancy and of 9.03 postpartum. The growing body of research on depression shows that depression and anxiety are often comorbid [51,52]. However, the extent to which depression or depressive symptoms impact the likelihood of anxiety is less clear. This study helps clarify these associations and underlines the need to assess the presence of both throughout the perinatal period. While directionality was not established here, the results suggest that depressive symptoms might exacerbate feelings of anxiety, particularly in women where depression is stigmatized and not properly addressed. Results from a previous study show that prenatal anxiety symptoms were predictive of postpartum depressive symptoms in Mexican women [10]. As can be seen in Table 1, 20 percent of prenatal and postpartum women had depressive symptoms, which is at the top end of the range among women in the general U.S. population [53] and other countries [54], yet within the range of Latinas in the U.S. at 20–43% [15,18,55,56,57]. This highlights the importance of early prevention and detection.

Unlike the previously discussed risk factors, adherence to the TFR was significantly associated with PAS in pregnancy. A recent systematic review showed that the TFR was inconsistently associated with prenatal depression, but significantly and indirectly associated with postpartum depression [58]. Although this is the first study to examine the associations between TFR and PAS, we can draw on previous literature, which suggests that the limitations and gender-related biases women experience, particularly in a male-dominated culture, induce anxious symptoms and increase the risk of other mental health issues [59]. Because of the male-dominated culture in Mexico and the submissive role women and mothers are expected to assume, it will be important to determine how and why adherence to the TFR increases the risk of anxiety in women in Mexico to inform interventions and target specific policies that perpetuate those practices [60]. An additional necessary step is to determine why the TFR increased the odds of anxiety in pregnancy in this sample of Mexican women. One possible explanation is the social pressures pregnant women in Mexico experience. As previously mentioned, motherhood is revered in Mexico, but so are behaviors that may dictate what a woman can or cannot do, such as only seeking mental health treatment or participating in a study interview or experiment after receiving approval from her partner [37] and prioritizing bearing children even though they are working women [61,62,63]. The TFR might also conflict with the growing independence observed in Mexican women. For instance, although women in Mexico are engaged in the labor force and pursue higher education, they continue to experience gender-role expectations, such as serving as the primary caregiver and being submissive to a male partner [62,63]. These gender-role restrictions and expectations may be heightened in pregnancy when motherhood is added to the mix. Given the potentially complex factors that contribute to the TFR, future studies should determine whether the effects of this risk factor differ from pre-conception to pregnancy to help advance theory and inform intervention.

Though this study addresses significant gaps in the literature, several limitations exist. Preconception data on the risk factors involved could help explain whether the effect of stressful life events and low social support on anxiety symptoms is a function of motherhood or a consequence of those events. Previous literature has shown that stressful life events are indirectly associated with anxiety in adolescence and adulthood because of the rumination that ensues in the wake of these events [64]. It is possible that the onset of motherhood may trigger feelings associated with these stressful life events, which in turn could increase the risk of anxiety in women. However, this is speculation that should be tested by including those who have not conceived or given birth. A second limitation is the expectations and effectiveness of social support. Studies on depression in the perinatal period have shown that women in the perinatal period have specific expectations of and needs for social support during this life transition [65,66,67]. However, less is known about expectations of social support and their effects on anxiety. Future studies should therefore explore this area of inquiry to inform interventions. Moreover, because effective support is important, subsequent studies should also inquire as to how women define the support, they deem efficacious and help them identify sources of support that can provide this. A third limitation is the lack of information regarding the TFR before gestation. As suggested above, motherhood may increase specific gender-role expectations in the woman and those around her. It will therefore be important to assess levels of adherence to the TFR before gestation. On a related note, future studies should inquire as to whether adherence to the TFR is self-imposed or influenced by those around a woman, which can help inform interventions and psychoeducation before gestation and throughout the perinatal period. Furthermore, women in this study were predominantly Mexican women living in Mexico. Thus, the findings may be limited to Mexican women and it is unclear whether the findings would be the same for women from other Latin American countries or Latinas in the U.S.

Despite these limitations, the findings from this study can help inform clinical practice and increase early diagnoses by health care providers, particularly in Mexico, which can increase early intervention and decrease the long-term effects of PAS. For example, health providers working with perinatal women should assess anxiety levels as well as the presence of the risk factors identified here. Given the multigenerational implications of PAS, women and children will benefit from early prevention and appropriate treatment in the short- and long-term.

## 5. Conclusions

This study is the first to investigate PAS rates among women living in Mexico. PAS risk factors such as stressful life events, low social support, and depression support the findings in the literature. However, our study also found that the traditional female role was a risk factor for anxiety in pregnancy. This finding shows that the traditional female role is a temporary and culturally relevant risk factor among these women. Early culturally appropriate interventions are therefore needed. Although there is a dearth of interventions designed to address gender-role risk factors among Mexican mothers [10,68], researchers and clinical practitioners can draw on culturally-relevant interventions designed to improve poor maternal mental health outcomes among women of Mexican descent [69,70,71] and other Latina women. Recent studies using complementary and integrative health interventions, such as music therapy [72,73], have yielded encouraging results. The use of complementary and integrative health approaches is particularly important given potential fears of stigma and barriers to traditional psychotherapeutic and pharmacological interventions among Latina women [74]. While our study sampled perinatal women living in Mexico, the findings suggest that future studies on women of Mexican descent should consider evaluating culturally relevant factors such as the traditional female role.

## Figures and Tables

**Figure 1 ijerph-18-00082-f001:**
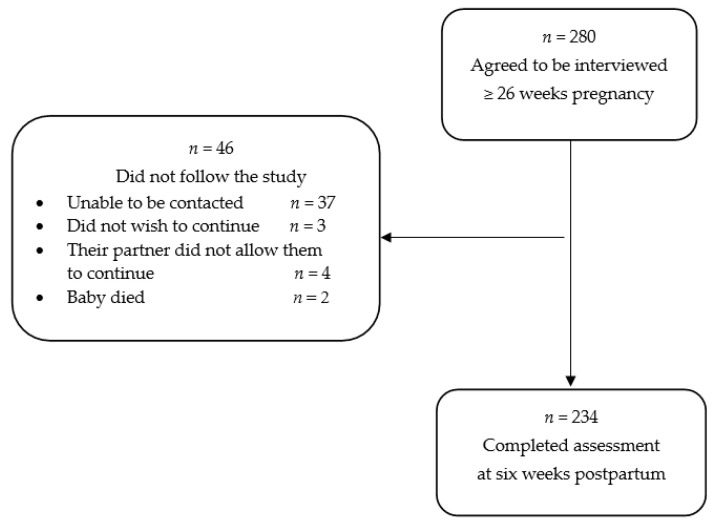
Participant flow diagram.

**Table 1 ijerph-18-00082-t001:** Demographic, obstetric, and psychosocial risk factor characteristics.

		$\bar{x}$	SD
Age		28.9	6.2
Monthly family income		498.7	502.3
Educational attainment (years)		12.7	3.8
Number of children		0.81	0.89
		*n*	%
Marital status	Partnered	223	79.6
	Single	57	20.4
Delivery	Vaginal	85	36.3
	Cesarean	149	63.7
Resilience during pregnancy	Low	57	20.4
	High	223	79.6
Traditional female role during pregnancy	Low	206	73.6
	High	74	26.4
Marital satisfaction during pregnancy	Low	68	24.3
	High	212	75.7
Social support during pregnancy	Low	42	15.0
	High	238	85.0
Stressful life events during pregnancy	With events	59	21.1
	Without events	221	78.9
Depressive symptoms during pregnancy	With symptoms	57	20.4
	Without symptoms	223	79.6
Social support during the postpartum period	Low	40	17.1
	High	194	82.9
Stressful life events during the postpartum period	With events	45	16.1
	Without events	189	67.5
Depressive symptoms during the postpartum period	With symptoms	40	20.1
	Without symptoms	194	79.9

Note: *n* = 234 women; $\bar{x}$ = Mean; SD = Standard Deviation.

**Table 2 ijerph-18-00082-t002:** Perinatal anxiety symptom rates.

	Pregnancy (*n* = 280)	Sixth Week Postpartum (*n* = 234)
	*n* (%)	*n* (%)
With anxiety symptoms	62 (22.1)	41 (17.5)
Without anxiety symptoms	218 (77.9)	193 (82.5)

McNemar χ^2^ = 1.30, *p* = 0.25.

**Table 3 ijerph-18-00082-t003:** Bivariate odds ratios (ORs) with 95% confidence intervals (CIs) for demographic, obstetric and psychosocial variables associated withanxiety symptoms in pregnancy and sixth postpartum week.

Variables	Time 1. Pregnancy (*n* = 280)	Time 2. Sixth Week Postpartum (*n* = 234)
	OR (95% CI)	OR (95% CI)
Age	0.99 (0.95–1.04)	0.97 (0.92–1.03)
Monthly family income	0.99 (0.89–1.03)	0.99 (0.84–1.49)
Single	2.59 (1.37–4.90) *	2.25 (1.05–4.80) *
Educational attainment	0.95 (0.89–1.03)	0.89 (0.81–0.97) *
Number of children	1.10 (0.81–1.50)	1.31 (0.78–1.63)
Cesarean delivery	--	1.15 (0.57–2.29)
Low resilience during pregnancy	2.87 (1.52–5.41) **	2.24 (1.03–4.89) *
Traditional female role during pregnancy	4.07 (2.23–7.41) **	1.95 (0.95–4.01)
Low marital satisfaction during pregnancy	4.03 (2.19–7.40) **	2.43 (1.17–2.42) *
Low social support during pregnancy	6.17 (3.07–12.41) **	6.63 (2.85–15.42) **
Stressful life events during pregnancy	8.40 (4.40–16.02) **	3.47 (1.66–7.27) **
Depressive symptoms during pregnancy	6.63 (3.49–12.58) **	4.55 (2.15–9.59) **
Low social support during the postpartum period	--	2.85 (1.31–6.18) *
Stressful life events during the postpartum period	--	7.39 (3.50–15.60) **
Depressive symptoms during the postpartum period	--	11.25 (5.14–24.61) **

** *p* ≤ 0.01, * *p* ≤ 0.05.

**Table 4 ijerph-18-00082-t004:** Multivariate logistic regression model of anxiety symptoms during pregnancy.

Pregnancy (*n* = 280)		
	OR	95% CI
Age	1.01	0.95–1.08
Monthly family income	0.99	0.93–1.10
Educational attainment	1.06	0.94–1.20
Number of children	0.85	1.50–6.68
Single	1.72	0.70–4.25
Low resilience during pregnancy	1.16	0.48–2.78
Traditional female role during pregnancy	2.28 *	1.06–4.88
Low marital satisfaction during pregnancy	1.32	0.54–3.25
Low social support during pregnancy	2.25	0.92–5.50
Stressful life events during pregnancy	4.35 **	2.02–9.37
Depressive symptoms during pregnancy	3.82 **	1.72–8.48

χ^2^ = 77.11 *p* = 0.00 ** *p* ≤ 0.01, * *p* ≤ 0.05.

**Table 5 ijerph-18-00082-t005:** Multivariate logistic regression model of six weeks postpartum anxiety symptoms.

Six Weeks Postpartum (*n* = 234)		
	OR	95% CI
Age	0.94	0.86–1.03
Monthly family income	0.99	0.96–1.22
Educational attainment	0.84 *	0.72–0.99
Number of children	0.82	0.32–2.11
Single	3.03	0.92–9.80
Cesarean delivery section	1.01	0.37–2.68
Low resilience during pregnancy	2.00	0.65–6.11
Traditional female role during pregnancy	0.90	0.31–3.08
Low marital satisfaction during pregnancy	5.06	1.50–20.84
Low social support during pregnancy	2.25 *	0.92–5.50
Stressful life events during pregnancy	0.98	0.31–3.08
Depressive symptoms during pregnancy	1.89	0.66–5.37
Low social support during the postpartum period	0.30	0.07–1.19
Stressful life events during the postpartum period	5.42 **	1.84–15.89
Depressive symptoms during the postpartum period	9.03 **	3.24–15.12

χ^2^ = 72.11 *p* = 0.00 ** *p* ≤ 0.01, * *p* ≤ 0.05.

## Data Availability

The data presented in this study are available on requested from the corresponding author. The data can be available from the repository of the Ramon de la Fuente Muñiz National Institute of Psychiatry.

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
