# Peer review of "Perinatal Anxiety Symptoms: Rates and Risk Factors in Mexican Women"

_ijerph, 2020, doi:10.3390/ijerph18010082_

Round 1
Reviewer 1 Report
The authors have addressed my previous comments and have also utilized a logistic regression, which was recommended. I have no further comments.
Reviewer 2 Report
The authors addressed all raised points, I have no further comments.
This manuscript is a resubmission of an earlier submission. The following is a list of the peer review reports and author responses from that submission.
Round 1
Reviewer 1 Report
The authors investigated the rates of perinatal anxiety symptoms (PAS) in Mexican women using a longitudinal study design. They found that PAS was prevalent in 21% in pregnant women and 18% postpartum. They also found that stressful events, poor social support, and depressive symptoms were associated with greater probability of PAS. This study is well-designed and much needed addition to the literature as this is understudied. I have some smaller concerns, and a larger one that needs to be addressed regarding the analysis. They utilized stepwise regression, which is known to have highly inflated p-values and regression coefficients if used without cross-validation.
- Stepwise regression (in fact all subset regression approaches) often have highly inflated r2 values(1, 2). Other approaches have been utilized that do not have as inflated r2 values (e.g., elastic net regression), however in both approaches it is typically necessary to conduct cross-validation to ensure that the data is not being overfit. In fact, this problem is well-documented especially for subset regression approaches like forward selection(3). Otherwise, the authors should potentially include all of these factors together in a single model, check that their variance inflation factors are below 5, and other model assumptions are met.
- Was there a reason to use cutoff points for age, education, median income and other measures?
- The SCL-90 is written in English, was there a translated version that was used? Could the authors comment on how cultural differences (especially given that SCL-90 was developed for English speaking individuals) may affect interpretation of reporting and descriptions of symptoms?
- The authors should help discuss potential interventions or programs that may help alleviate some of these stressors (e.g., TFR, low social support, and depression).
References
- Berk KN: Comparing subset regression procedures. Technometrics 1978; 20:1-6
- Rencher AC,Pun FC: Inflation of R2 in best subset regression. Technometrics 1980; 22:49-53
- Copas JB: Regression, prediction and shrinkage. Journal of the Royal Statistical Society: Series B (Methodological) 1983; 45:311-335
Author Response
The authors investigated the rates of perinatal anxiety symptoms (PAS) in Mexican women using a longitudinal study design. They found that PAS was prevalent in 21% in pregnant women and 18% postpartum. They also found that stressful events, poor social support, and depressive symptoms were associated with greater probability of PAS. This study is well-designed and much needed addition to the literature as this is understudied. I have some smaller concerns, and a larger one that needs to be addressed regarding the analysis. They utilized stepwise regression, which is known to have highly inflated p-values and regression coefficients if used without cross-validation.
Response: As described below, we modified the analysis according to your observations and added the missing information.
- Stepwise regression (in fact all subset regression approaches) often have highly inflated r2values(1, 2). Other approaches have been utilized that do not have as inflated r2values (e.g., elastic net regression), however in both approaches it is typically necessary to conduct cross-validation to ensure that the data is not being overfit. In fact, this problem is well-documented especially for subset regression approaches like forward selection(3). Otherwise, the authors should potentially include all of these factors together in a single model, check that their variance inflation factors are below 5, and other model assumptions are met.
Response: Thank you for providing the relevant references.
Although several statisticians do not recommend the stepwise method as a suitable strategy for evaluating the impact of risk factors on a dependent variable, some researchers continue to use it as there is a theoretical framework that can support the resulting models using this method.
However, after careful analysis of the recommended literature, we agreed with your observation and conducted multivariate analyses reporting all the variables included in the model.
Similar results to those reported previously were found with the new models.
See lines: 144-145; Table 1; 150-152; Table 3;182-183; Table 4; 202; 204; 206-208; Table 5; 210-221; 226-227; 238-239;248
- Was there a reason to use cutoff points for age, education, median income and other measures?
Response: In the initial analyses, the sociodemographic variables were dichotomized to facilitate their interpretation in the logistic regression and because the literature has shown that low educational attainment, lower age and lower income are predictors of perinatal depression and anxiety.
However, since we know that from a statistical point of view, transforming interval variables into dichotomous ones reduces the power of the variable, bivariate and multivariate regressions were carried out with non-dichotomized sociodemographic variables, showing results similar to the ones reported previously. Now the sociodemographic variables are reported at an interval measurement level.
- The SCL-90 is written in English, was there a translated version that was used? Could the authors comment on how cultural differences (especially given that SCL-90 was developed for English speaking individuals) may affect interpretation of reporting and descriptions of symptoms?
Response: We have now included the references on the validation of the SCL-90 in Mexico. The scale has been extensively used in Mexico and no-one has mentioned any aspect of concern when it has been used in this population (please see line 86)
- The authors should help discuss potential interventions or programs that may help alleviate some of these stressors (e.g., TFR, low social support, and depression).
Response: We thank the reviewer for their suggestion. We added recommendations for culturally appropriate interventions in the conclusions (please see verses 318-325)
Reviewer 2 Report
Dear Authors,
The presented study tackles a very importatnt issue of Perinatal Anxiety Symptoms: Rates and Risk Factors in Mexican Women. The study was conducted in an environment which is specific in terms of the culture, which constitutes an additional value of the paper .The study was conducted reliably with appropriate selection of tests. Overall, I think that this article should be published.
However, one issues require complementary information:
1. Verse 79 I suggest including the number of of the Institutional Review Board approval
2. Wers 87 Is the income in w American Pesos or in US Dollars? It’s not clear to me?- I suggest using USD
3. Verse 300-314 I would consider including: Zareba et al. Role of Social and Informational Support while Deciding on Pregnancy Termination for Medical Reasons
4. Many bibliographies are a little obsolete. The authors must update the bibliography ( 2019 and 2020) e.g.
• Zanardo et al. Maternity blues: a risk factor for anhedonia, anxiety, and depression components of Edinburgh Postnatal Depression Scale.
• Chen et al. A comparative study of domestic decision-making power and social support as predictors of postpartum depressive and physical symptoms between immigrant and native-born women
• Banasiewicz et al. Perinatal Predictors of Postpartum Depression: Results of a Retrospective Comparative Study.
• Prediction of postpartum depression based on women’s quality of life. fmpcr 2019, 21, 343–348, doi:10.5114/fmpcr.2019.90165.
Or you can see the attachment.

Author Response
- Verse 79 I suggest including the number of the Institutional Review Board approval
Rsponse: Due to an IRB error, the parent project approval document was not given a number. However, this has not been an impediment to publish the data of this study.
- Lara, M.A., Navarrete, L., Nieto, L., Barba, M.J.P, Navarro J.L, Lara-Tapia H. (2015). Prevalence and incidence of perinatal depression and depressive symptoms among Mexican women. Journal of Affective Disorders,175:18 24.http://dx.doi.org/10.1016/j.jad.2014.12.035
- Navarrete, L., Lara, M.A., Nieto, L., Lara, M.C. (2019). Sensitivity and Specificity of the Whooley Questions for Perinatal Depression in Mexican Women. Salud Pública de México, 61(1), 27-34. DOI> 10.21149/9083
- Wers 87 Is the income in w American Pesos or in US Dollars? It’s not clear to me?- I suggest using USD
Response: The income was originally in Mexican pesos but we converted it to the equivalent in US dollars because this would be easier to understand for non-Mexicans. It corresponds to approximately $478.70 USD per month please see line 150
- Verse 300-314 I would consider including: Zareba et al. Role of Social and Informational Support while Deciding on Pregnancy Termination for Medical Reasons
Response: The reference has been included in the line 293
- Many bibliographies are a little obsolete. The authors must update the bibliography ( 2019 and 2020) e.g.
- Zanardo et al. Maternity blues: a risk factor for anhedonia, anxiety, and depression components of Edinburgh Postnatal Depression Scale.
- Chen et al. A comparative study of domestic decision-making power and social support as predictors of postpartum depressive and physical symptoms between immigrant and native-born women
- Banasiewicz et al. Perinatal Predictors of Postpartum Depression: Results of a Retrospective Comparative Study.
- Prediction of postpartum depression based on women’s quality of life. fmpcr 2019, 21, 343–348, doi:10.5114/fmpcr.2019.90165.
Response: Thank you for the observation; we have updated the bibliography to reflect recent studies.
See lines: 31,35,40-41,43-44, 228, 249, 257-258,263, 266 and 293
We were not able to include two of the suggested articles:
- Banasiewicz et al. Perinatal Predictors of Postpartum Depression: Results of a Retrospective Comparative Study.
- Prediction of postpartum depression based on women’s quality of life. fmpcr 2019, 21, 343–348, doi:10.5114/fmpcr.2019.90165.
The topic of perinatal depression and anxiety has been insufficiently explored in Mexico. As a result, we cite the available research, which may not be as recent
Reviewer 3 Report
The authors conducted a secondary data analysis of perinatal anxiety symptoms (during pregnancy and at six weeks post-partum) in a sample of 234 Mexican women from a longitudinal study of perinatal depression in Mexico. The authors aimed to describe rates of perinatal anxiety symptoms as well as to identify risk factors, with a specific focus on adherence to the traditional female role, which was found to be associated with increased risk of anxiety symptoms. The topic is of high interest, the manuscript is well written and the results are relevant. There are some methodological aspects that might benefit of improvement.
- The authors should provide a strong rationale for the decision to dichotomize each quantitative variable. For instance, why age was not analyzed as a continuous variable and a cut-off at 28 years had to be identified?
- The authors conducted a logistic regression using a stepwise variable selection method approach. This approach is quite criticized because of its limitations, one of the main is that remaining coefficients are biased and need shrinkages. I would suggest either not to use authomated methods for variable selection (i.e. include all variables in the model) or, if the authors want to use them, to use methods alternative to stepwise regression. See, for instance, some documentation on problems related to stepwise regression:
https://journalofbigdata.springeropen.com/articles/10.1186/s40537-018-0143-6
https://www.stata.com/support/faqs/statistics/stepwise-regression-problems/
https://stats.stackexchange.com/questions/29851/does-a-stepwise-approach-produce-the-highest-r2-model
- The sentence at page 1, lines 32-33 repeats what was already stated in the first two sentences
- At page 3, line 79, the name of the Institution whose IRB approved the study has been removed for blind review, but authors names and affiliations are visible
- At page 4, line 161, in the sentence "Almost half of the participants were 28 years old or younger1", what does the 1 stands for?
Author Response
- The authors should provide a strong rationale for the decision to dichotomize each quantitative variable. For instance, why age was not analyzed as a continuous variable and a cut-off at 28 years had to be identified?
Response: This issue has been addressed in Reviewer 1, points 1 and 2.
- The authors conducted a logistic regression using a stepwise variable selection method approach. This approach is quite criticized because of its limitations, one of the main is that remaining coefficients are biased and need shrinkages. I would suggest either not to use authomated methods for variable selection (i.e. include all variables in the model) or, if the authors want to use them, to use methods alternative to stepwise regression. See, for instance, some documentation on problems related to stepwise regression:
https://journalofbigdata.springeropen.com/articles/10.1186/s40537-018-0143-6
https://www.stata.com/support/faqs/statistics/stepwise-regression-problems/
https://stats.stackexchange.com/questions/29851/does-a-stepwise-approach-produce-the-highest-r2-model
Response: This issue has been addressed in Reviewer 1, points 1 and 2.
- The sentence at page 1, lines 32-33 repeats what was already stated in the first two sentences
Response: It was removed to avoid repeating the information included in the first two sentences. Please see line 31
- At page 3, line 79, the name of the Institution whose IRB approved the study has been removed for blind review, but authors names and affiliations are visible
Response: Thank you for the observation. We have removed “For blind review” and have added the name of the institution that approved the study. Please see line 78.
- At page 4, line 161, in the sentence "Almost half of the participants were 28 years old or younger1", what does the 1 stands for?
Response: Thank you for the observation. We removed number 1, which was a mistake.